# Generalisation and the Geometry of Class Separability

**Dominic Belcher**
Department of Electronics
Computer Science
University of Southampton
Southampton, UK
db7g17@soton.ac.uk

**Adam Prugel-Bennett**
Department of Electronics
Computer Science
University of Southampton
Southampton, UK
apb@ecs.soton.ac.uk

**Srinandan Dasmahapatra**
Department of Electronics
Computer Science
University of Southampton
Southampton, UK
sd@ecs.soton.ac.uk

## Abstract

Recent results in deep learning show that considering only the capacity of machines does not adequately explain the generalisation performance we can observe [7, 11]. We propose that by considering the geometry of the data we can better explain generalisation achieved in deep learning. In particular we show that in classification the separability of the data can explain how good generalisation can be achieved in high dimensions. Further we show that layers within a CNNs sequentially increase the linear separability of data, and that the information these layers retain or discard can help explain why these models generalise.

## 1 Introduction

Existing work studying generalisation in machine learning has typically focused on the properties of learning machines, most significantly on their capacity and the size of the hypothesis class [4, 6, 9]. Relatively little work has explored the geometry of the data, and how this impacts the generalisation performance in different settings. Empirical results show that the capacity of a machine is not by itself a good indicator of generalisation performance [7], and therefore focusing only on the hypothesis class does not give the full picture of generalisation in a given setting. It is therefore important to investigate how the data impacts generalisation.

In many cases deep learning is applied to data which live in a large number of dimensions [2], in which it is difficult to make sense of its geometry. A key observation is that the distance between data points has little meaning in high dimensions, since this grows with the square root of the number of dimensions, making similar data points appear distant [1].

We hypothesise that in many different settings our real data lives on a low dimensional manifold within the data space, and that the geometry of this manifold is directly relevant to both the suitability of different machines and the generalisation performance these can obtain. Moreover we believe that how the geometry of this manifold is transformed by a machine is key to understanding the information captured by the machine, and therefore also to understanding how certain machines are

34th Conference on Neural Information Processing Systems (NeurIPS 2020), Vancouver, Canada.

able to generalise. This is of particular significance in the case of Deep Learning, in which data is successively transformed by many functions to yield the output [2].

## 2 Our work

We investigate projecting the data into a low dimensional subspace in order to give insight into its geometry, in particular in classification, where we aim to find a projection which captures the specific geometry of class separation. We also present the idea that in the case of classification the separability of the data has an important role in generalisation, and show that the separation of classes can yield good generalisation even in high dimensions.

### 2.1 Theoretical results on class separation

We consider the case where we have data consisting of $p$-dimensional feature vectors $\boldsymbol{X}$ and the task is to classify them into two classes $\pm 1$. We assume the data points, $(\boldsymbol{X}, Y)$, are distributed according to $p_{\boldsymbol{X}, Y}(\boldsymbol{x}, y) = p_{\boldsymbol{X}|Y}(\boldsymbol{x} \mid y) \, \mathbb{P}_Y[Y = y]$

$$p_{\boldsymbol{X}|Y}(\boldsymbol{x} \mid y) = \mathcal{N}(\boldsymbol{x} \mid y \, \Delta \, \boldsymbol{x}^*, \mathsf{I}_p) \qquad \mathbb{P}_Y[Y = y] = \frac{1}{2} \left( [\![y = 1]\!] + [\![y = -1]\!] \right) \tag{1}$$

where $[\![\text{predicate}]\!]$ is an indicator function, $\Delta$ is the distance along some $\boldsymbol{x}^* \in \mathbb{R}^p$ from the mean of each class to the origin, such that the means are separated by $2\Delta$, and $|\boldsymbol{x}^*| = 1$. We assume that we have a training data set, $\mathcal{D}$, consisting of $m$ examples drawn independently from $p_{\boldsymbol{X}, Y}(\boldsymbol{x}, y)$.

We consider a perceptron with weight vectors $\boldsymbol{w}_h \in S^p$ (the $p$-sphere) that makes predictions $h(\boldsymbol{X}) = \text{sign}(\boldsymbol{w}_h^\mathsf{T} \boldsymbol{X})$. We define a loss function for a hypothesis $h$ (i.e. the perceptron with weights $\boldsymbol{w}_h$) to be

$$L_h(\boldsymbol{X}, Y) = [\![h(\boldsymbol{X}) \neq Y]\!]$$

The risk of a hypothesis, $h$, is then given by

$$R_h = \Phi(-\Delta \, \cos(\theta_h)) \tag{2}$$

where $\cos(\theta_h) = \boldsymbol{w}_h^\mathsf{T} \boldsymbol{x}^*$ and $\Phi(z)$ is the cumulative distribution function for a standard normal distribution..

We assume are are given $m$ training examples $\{(\boldsymbol{X}^\alpha, Y^\alpha) | \alpha = 1, 2, \ldots, m\}$ drawn independently from $p_{\boldsymbol{X}, Y}$. We consider a *Hebb classifier* where we choose a weight vector $\boldsymbol{w} = \bar{\boldsymbol{w}}/|\bar{\boldsymbol{w}}|$ where

$$\bar{\boldsymbol{w}} = \sum_{\alpha=1}^{m} Y^\alpha \, \boldsymbol{X}^\alpha. \tag{3}$$

Given that the risk of a weight vector with an angle of $\theta$ from $\boldsymbol{x}^*$ is $\Phi(-\Delta \, \cos(\theta))$ then the expected risk using the Hebb rule can be approximated as

$$\bar{R} \approx \bar{R}_{approx} = \Phi\left(-\frac{\Delta}{\sqrt{1 + \frac{p}{m \Delta^2}}}\right) \tag{4}$$

In Figure 1 we show the expected risk versus the number of training examples for feature vectors of length $p = 50, 100$ and $200$ and for different separations of the two distribution $\Delta = 1, 2, 3$ and $4$. In addition, we show simulation results averaged over 100 different training runs. Clearly, this approximation provides a very good approximation to the observed behaviour.

### 2.2 Empirical work on CNNs

Our work focuses on the application of CNNs in image classification. Image data exemplifies the low dimensional geometry of the true data which we are interested in, as is evidenced by the fact that randomly sampling the image space would never produce anything resembling a recognisable image. We can also show by means of a simple example that classes within image data are not linearly separable, as is shown in the separability of the raw data in figure 4.

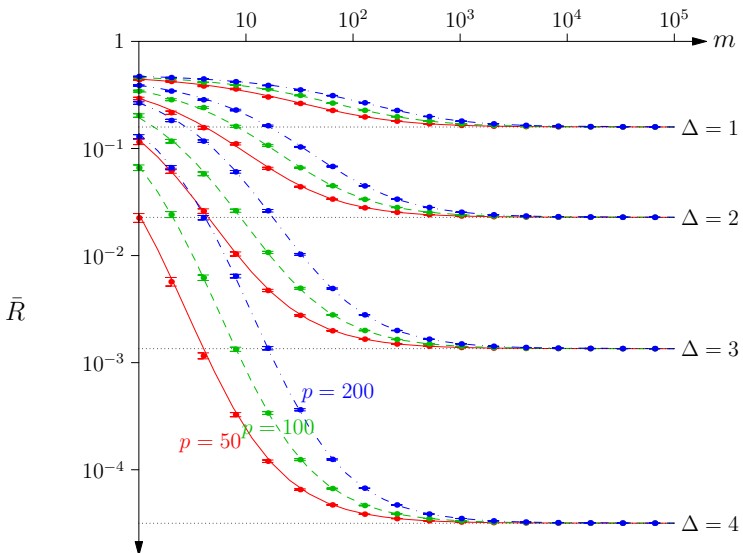

Figure 1: The expected risk of the simple learning machine versus the number of training examples, $m$. We also show simulation results averaged over 100 runs.

CNNs are of specific interest as they seem to be particularly well attuned for use on image data, as is clear from the multitude of recent work showcasing exceptional performance [5, 3, 10]. Indeed the state of the art performance on ImageNet is achieved by a CNN with 480 million parameters [8], exemplifying the point that capacity is not a good indicator for generalisation. We hypothesise that the geometry of real image data is directly related to the suitability of CNNs to images, and we believe that the way in which convolutional layers transform the data is of particular significance in understanding how CNNs achieve such exceptional generalisation performance.

As we have shown the significance of separability in generalisation, we investigate the linear separability of data as it is transformed by a CNN. In order to achieve good classification performance, the data must become linearly separable in the final layer, however there is no requirement for linear separability at any previous layer, due to the use of nonlinear activations. Nevertheless, increasing linear separability would indicate that the convolutional layers transform the geometry of the data in a way that enables good generalisation.

We take a trained CNN $F : X \rightarrow Y$, and split it at some layer $l$, to give us a network $F_l : X \rightarrow Z_l$, with $F_l$ being the network given by the layers of $F$ up to and including $l$. We call $Z_l$ the *Representation Space* of $F_l$. We then map the data using $F_l$ to give us a re-representation of this data in the Representation Space. By projecting the data in each Representation Space into a low dimensional subspace, we can examine how salient information is extracted throughout the network, and spurious information is filtered out.

We then train a linear classifier on the re-represented data in the Representation Space, and take the test accuracy of the classifier to be an estimate of the relative linear separability of the data. In a C-class problem a linear classifier is simply a projection into C dimensions, which aims to capture as much salient information as possible.

We experiment using CNNs of varying depths trained on CIFAR10. We term an $N$ layer CNN to be a CNN with $N$ convolutional layers. For the purposes of splitting the network we consider each operation, such as a convolution, activation, or max pooling, to be its own layer, as we are interested in how each of these transform the data.

## 3  Results

We look at three CNN models, of 4, 6 and 9 convolutional layers. Figure 2 shows the percentage change in separability in the 9 layer model between each layer, measured on the training and test

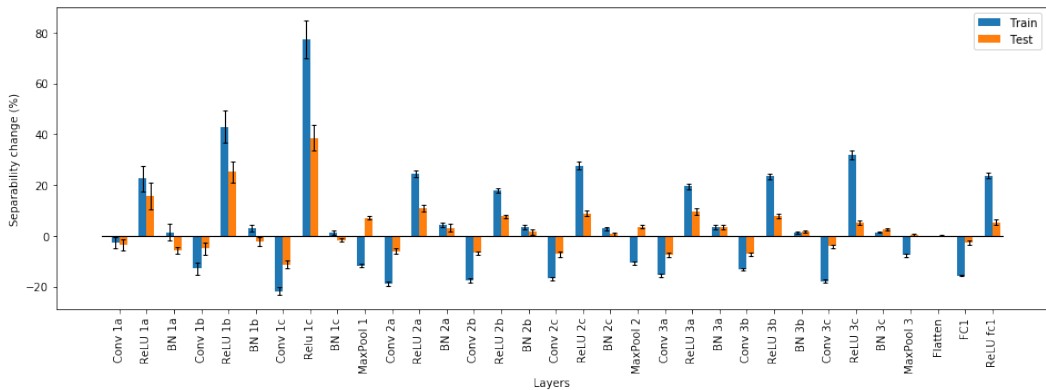

Figure 2: Change in separability of data given by gradient descent

data. Each layer corresponds to the change between that layer and the one immediately preceeding it. These results are averaged over 15 trials. See appendix A for results from all models.

Our results demonstrate a clear trend of the linear separability increasing throughout the network, both on the training and test data, giving a clear indication that the geometry of the data is becoming more separated. We observe that this increase is not monotonic, especially in the training data separability.

There is a clear trend mirrored across each network of the separability decreasing with each convolution, followed by a sharp increase following the activation function. We see this trend in both the training and test data, however it is much more pronounced in the training data, which may be indicative of overfitting, in that the network is extracting some spurious information on the training data as well as salient information on the underlying distribution. It seems clear that the convolution and activation work in tandem to augment the data in such a way as to make it more separable, thereby giving improved performance, but that this can overfit to spurious information within the training data.

In contrast, we also observe that the max pooling layers induce significant decreases in the separability of the training data, but increases in separability of the test data. This suggests that the max pooling layers help to prevent overfitting, by discarding spurious information but retaining salient information. We can interpret max pooling layers as focusing on strong responses to filters, indicating the presence of particular features, but discarding fine-grained positional information. This is somewhat unsurprising, as the position of an object is not relevant for classifying it. This suggests that max pooling plays an important role in transforming the geometry of the data to improve generalisation.

Our results are a strong indication that linear separability emerges throughout the subsequent layers of a CNN, and we believe this to be an important factor in the exceptional generalisation performance these models can achieve.

## 4 Conclusions

Our results show that the geometry of class separation in data is significant in the generalisation performance which can be obtained in learning. Moreover our results show how seperability emerges in sequential layers of a trained deep network, giving insight into why deep networks are able to generalise despite over-parameterisation. Specifically we have shown that particular operators in a network contribute to this in different ways, in particular that max pooling has an important property of distilling salient information from data, giving improved separability. We believe the hierarchical construction of complex features exhibited by deep networks is directly related to the process of layers sequentially filtering spurious information from data, which prevents deep networks overfitting and gives good generalisation. We believe linear separability in the representation of the data to be characteristic of this, as our results show linear separability to be both important for generalisaiton, and evident in deep networks.

## Broader Impact

We believe this to not be applicable to our work.

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

# Appendix

## A  Full Results

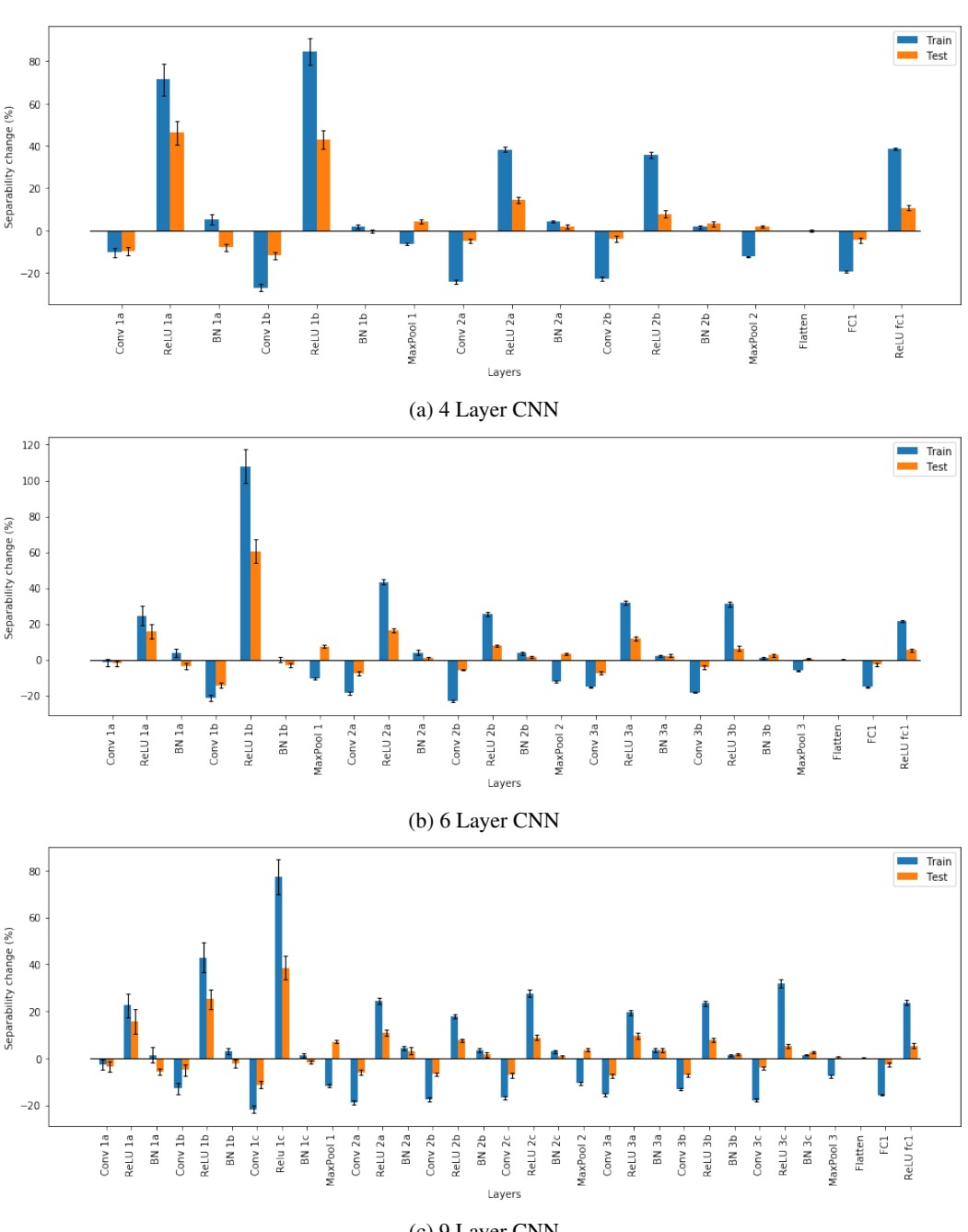

(a) 4 Layer CNN

(b) 6 Layer CNN

(c) 9 Layer CNN

Figure 3: Change in separability of data between layers

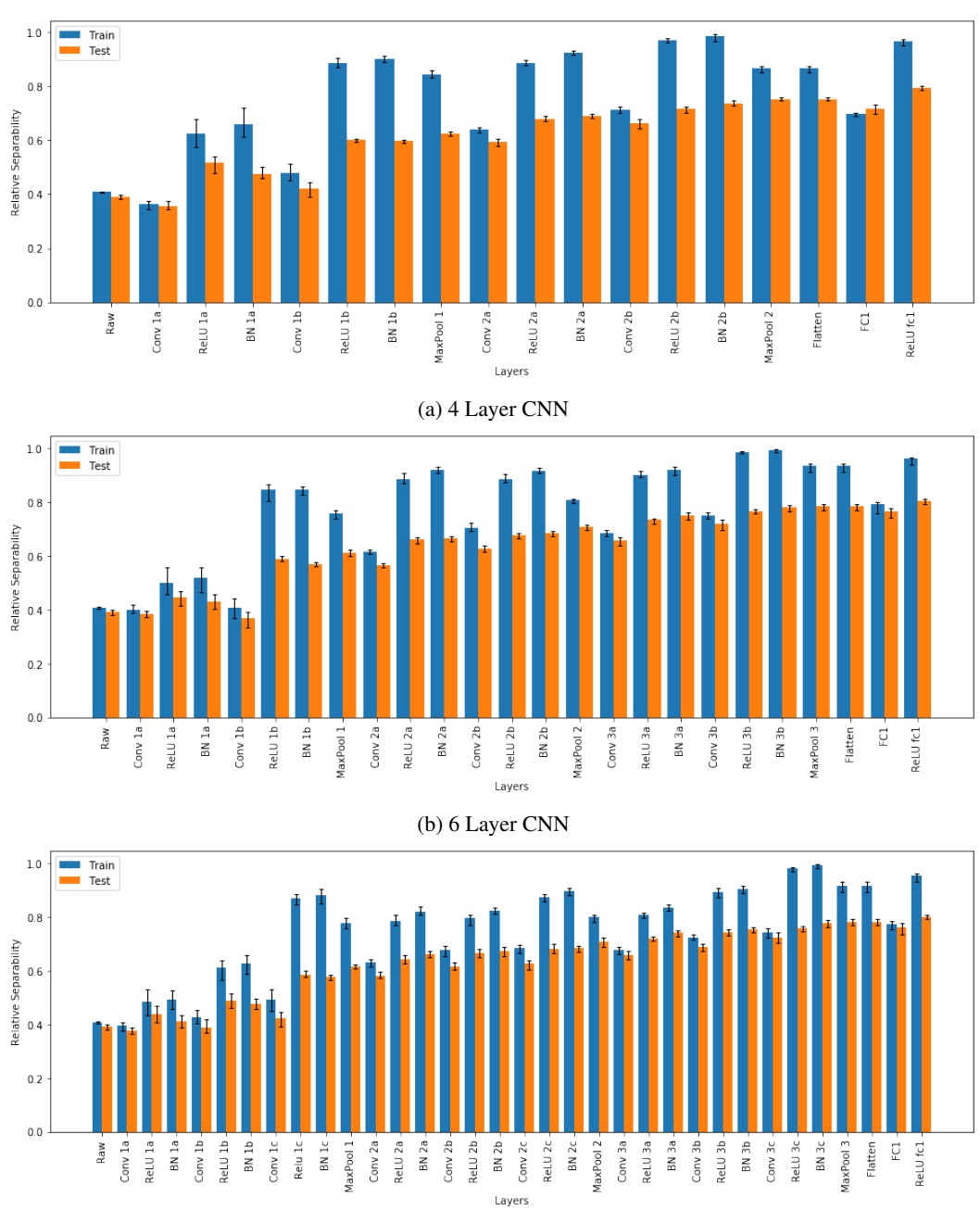

(a) 4 Layer CNN

(b) 6 Layer CNN

(c) 9 Layer CNN

Figure 4: Relative separability of data at each layer

