# OpenReview forum: "Generalisation and the Geometry of Class Separability"
_NeurIPS.cc/2020/Workshop/DL-IG — NeurIPSW 2020: DL-IG Poster_

### Official Review · AnonReviewer2 · 2020-10-31
**Linear separability in layers.**

**Rating:** 6
**Confidence:** 3

**Review:**

This work investigates how linearly separable the features in a network are as a function of network depth.  I think looking at linear separability is a bit of a naive probe to apply in this case.

Instead of taking the linear classifier accuracy as a measure of the linear separability, why not instead interpret the linear classifiers likelihood as a variational lower bound on the mutual information between the labels and the layers representation?  You could then go beyond linear and measure how much information (or a variational lower bound there of) of how much information is present at each layer of the network, which would likely be of more interest to the crowd of this workshop.

There is related work that I think goes a bit further, including this older work: https://arxiv.org/abs/1905.00414 and its more recent followup (which just came out so this paper can't have hoped to have referenced, but I include it anyway because the authors should find it interesting: https://arxiv.org/abs/2010.15327

Overall, I think this is find for the workshop, but the work would probably need more indepth experiments if it were to be expanded into a full paper.

---

### Official Review · AnonReviewer1 · 2020-11-07

**Rating:** 6
**Confidence:** 4

**Review:**

This paper attempts to explain the generalization in a deep network using separability of the features at various layers. The paper is motivated via a perceptron classifying a Gaussian distribution.

Overall, I think the experimental evidence in the paper is not rigorous enough to draw the conclusions that the authors wish to draw. The paper will benefit from more careful experiments, I am listing down some suggestions below.

What is the connection between the result in Section 2 with those in Section 3? What is the variance on the plots in Fig. 2?
On lines 103-110, max-pooling should reduce variability because it collapses spatial variability in the input. I think there is confounding occurring here due to the presence of batch-norm which (at least in theory) de-correlates the features. De-correlated features are easier to separate using a linear classifier. Further, the features of well-performing deep networks are not easily separable, e.g., https://arxiv.org/abs/1902.01889.

---

### Author Response · Authors · 2020-12-12
**Video Presentation**

Youtube link: https://youtu.be/uEwLG7VCYdM

---

### Decision · Program_Chairs · 2020-11-07

**Decision:**

Accept (Poster)

**Comment:**

Please incorporate the reviewer feedback into your manuscript.